# Approximating CKY with Transformers

**Ghazal Khalighinejad**
Duke University
gk126@duke.edu

**Ollie Liu**
University of Southern California
zliu2898@usc.edu

**Sam Wiseman**
Duke University
swiseman@cs.duke.edu

## Abstract

We investigate the ability of transformer models to approximate the CKY algorithm, using them to directly predict a sentence's parse and thus avoid the CKY algorithm's cubic dependence on sentence length. We find that on standard constituency parsing benchmarks this approach achieves competitive or better performance than comparable parsers that make use of CKY, while being faster. We also evaluate the viability of this approach for parsing under *random* PCFGs. Here we find that performance declines as the grammar becomes more ambiguous, suggesting that the transformer is not fully capturing the CKY computation. However, we also find that incorporating additional inductive bias is helpful, and we propose a novel approach that makes use of gradients with respect to chart representations in predicting the parse, in analogy with the CKY algorithm being a subgradient of a partition function variant with respect to the chart.

## 1 Introduction

Parsers based on transformers (Vaswani et al., 2017) currently represent the state of the art in constituency parsing (Mrini et al., 2020; Tian et al., 2020), and recent work (Tenney et al., 2019; Jawahar et al., 2019; Li et al., 2020; Murty et al., 2022; Eisape et al., 2022; Zhao et al., 2023) has found that transformers are capable of learning constituent-like representations of spans of text. Given these successes, it is natural to wonder whether transformers capture the algorithmic processes we associate with constituency parsing, such as the CKY algorithm (Kasami, 1966; Younger, 1967; Baker, 1979). Indeed, one might suspect that the layers of a transformer are building up phrase-level representations, much as the CKY algorithm itself builds up its chart. Such a hypothesis has become particularly compelling in light of recent work studying the ability of transformers, and graph neural

networks more generally, to implement or approximate classical algorithms (Xu et al., 2019; Csordás et al., 2021; Dudzik and Veličković, 2022; Delétang et al., 2022, *inter alia*).

If standard transformers were indeed approximating CKY, there would be several implications. Practically, such a finding might lead to faster neural parsers. Whereas state-of-the-art parsers (e.g., that of Kitaev and Klein (2018) and follow-up work (Kitaev et al., 2019; Tian et al., 2020)) tend to implement CKY on top of transformer-representations, thus incurring a parsing time-complexity that is cubic in the sentence length, we could conceivably extract a parse from simply running a transformer over the sentence; this would involve a time-complexity cost that is only quadratic in the sentence length. Moreover, since there is significant academic and industrial effort aimed at making standard transformers faster (e.g., that of Dao et al. (2022)), this progress could conceivably transfer automatically to the parsing case.

In addition to more practical considerations, the task of producing CKY parses with transformers provides an excellent opportunity for investigating whether endowing transformers with additional inductive bias can help them in implementing classical algorithms, a topic recently studied by Csordás et al. (2021) and others. The results of such an investigation would also bear on recent results relating to the computational power of transformers trained in the standard way (Delétang et al., 2022; Liu et al., 2023).

Accordingly, we first show that having a pretrained transformer simply predict an entire parse by independently labeling each span — an approach similar to that taken in a different context by Corro (2020) only at *training time* — is sufficient to obtain competitive or better performance on standard constituency parsing benchmarks, while being significantly faster.

While these constituency parsing results are en-

couraging, they do not imply that trained transformers are implementing the CKY algorithm, because the transformer may simply be predicting a parse without it being highest-scoring under some grammar. We accordingly go on to investigate transformers' performance in predicting a CKY parse under *randomly generated* PCFGs. Given a randomly generated PCFG, it is of course easy to check whether a predicted parse is indeed highest-scoring. In this setting we find that the performance of transformers negatively correlates with the ambiguity of the PCFG, suggesting that they are not in fact implementing something CKY-like. At the same time, we find that incorporating additional inductive bias into the standard architecture is helpful, and we propose a novel approach, which makes use of *gradients* with respect to chart representations in predicting the parse. This inductive bias is inspired by the fact that the CKY algorithm can be viewed as computing the gradient of the "max score" partition function (Eisner, 2016; Rush, 2020), and we find that this improves performance on random PCFGs significantly.

In summary, we show that:

- using transformers to directly predict parses performs competitively with explicit CKY-based approaches, while being faster;

- this impressive performance is likely *not* due to transformers implicitly implementing the CKY algorithm, as they fail to accurately parse ambiguous synthetic PCFGs;

- biasing the model to produce parses from gradients with respect to the chart significantly improves synthetic PFCG parsing performance.

Code for reproducing all experiments is available at https://github.com/ghazalkhalighinejad/approximating-cky.

## 2 Background and Notation

The CKY algorithm (Kasami, 1966; Younger, 1967; Baker, 1979) computes a highest scoring parse of a sentence under a probabilistic context-free grammar (PCFG). Let $G = (\mathcal{N}, \Sigma, \mathcal{R}, S, \mathcal{W})$ be a PCFG, with $\mathcal{N}$ the set of non-terminals, $\Sigma$ the set of terminals, $\mathcal{R}$ the set of rules, $S$ a start non-terminal, and $\mathcal{W}$ a set of probabilities per rule (which normalize over left-hand-sides). Given a sentence $x \in \Sigma^T$, the CKY algorithm uses a dynamic program to compute a highest scoring parse

of $x$ under $G$, and it requires $O(|\mathcal{R}|T^3)$ computation time.

Following the notation in Rush (2020), let $\ell^R \in \mathbb{R}^{|\mathcal{R}| \times |\mathcal{N}| \times |\mathcal{N}|}$ represent the log potentials (e.g., log probabilities) associated with the rules in PCFG $G$, and let $\ell^E(x) \in \mathbb{R}^{T \times |\mathcal{N}|}$ represent the log potentials corresponding to each token in input sentence $x$. We can use these log potentials to compute the chart $\beta \in \mathbb{R}^{T \times T \times |\mathcal{N}|}$ for $x$, where $\beta[i, j, a]$ represents the sum (under a particular semiring) of all weight associated with the $a$-th non-terminal yielding $x_{i:j}$. These log potentials can also be used to compute a highest-scoring parse, which we will refer to as $\beta^* \in \{0, 1\}^{T \times T \times |\mathcal{N}|}$; this is the one-hot representation of a parse, with $\beta^*[i, j, a] = 1$ iff $j \geq i$ and nonterminal $a$ yields $x_{i:j}$ in the parse.

**CKY as a subgradient** Recent work (Eisner, 2016; Rush, 2020) has emphasized that inference algorithms such as CKY may be fruitfully viewed as a special case of calculating the gradient or a subgradient of a generalized log partition function with respect to (some function of) the log potentials associated with an input. In particular, the CKY algorithm can be viewed as computing a subgradient of the "max score" variant of the log partition function with respect to an input sentence $x$'s chart; see Appendix A for details. Thus, $\beta^*$ is relatively easily obtained with automatic differentiation frameworks. However, computing it in this way is exactly equivalent to running the traditional CKY algorithm, and so still requires $O(|\mathcal{R}|T^3)$ time.

**Transformer-based CKY approximations** The remainder of the paper focuses on training a model (i.e., an inference network (Kingma and Welling, 2014; Johnson et al., 2016; Tu and Gimpel, 2018)) to predict $\beta^*$ directly, in the hope of parsing more efficiently. Rather than use $\ell^R$ and $\ell^E(x)$ to recursively compute $\beta^*$ as the traditional CKY algorithm does, or compute $\beta^*$ by differentiating with respect to $\beta$, we instead propose to simply train a transformer to map a sentence to a correct $\beta^*$ for it. The hope is that the trained transformer will learn to represent information about the grammar implicitly in its weights, and that it can then be used without log potentials to parse unseen sentences from the same grammar. This approach is illustrated in schematic form in Figure 1.

We seek to evaluate our trained inference network's approximation to $\beta^*$ on held-out sentences that are *from the same distribution* (i.e., sampled

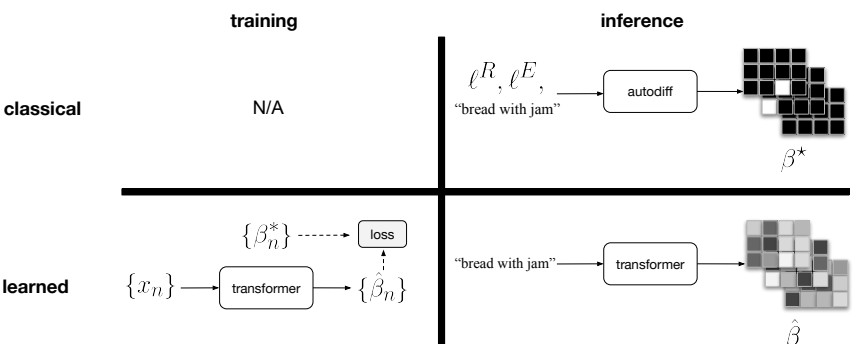

Figure 1: A schematic view of training and inference under the classical approach (top) and under learned transformer approximation (bottom). Classical parsing has no training phase, and uses pre-defined log potentials to parse unseen sentences. Our proposed learned parser (an "inference network") trains on pairs of sentences and parses computed under a set of log potentials, and then parses unseen sentences without access to the potentials.

from the same PCFG) as the training sentences. This contrasts with the vast majority of work on approximating classical algorithms with neural networks (Graves et al., 2014; Kaiser and Sutskever, 2016), which instead evaluates performance on out-of-distribution, and in particular *longer*, inputs than those on which the model was trained. We do not focus on length generalization, first because we believe generalizing even in-domain is useful in parsing applications, and second because we find transformers trained on random PCFGs struggle to generalize even to inputs of the same length.

**CKY variants**   Modern span-based neural constituency parsers do not assume an underlying PCFG. Rather, these parsers score spans compositionally (Stern et al., 2017; Gaddy et al., 2018; Kitaev and Klein, 2018) using log potentials $\ell^S \in \mathbb{R}^{T \times T \times |\mathcal{N}|}$ produced by a neural network, such as a transformer. Such parsers use a CKY variant that requires $O(T^3)$ computation time. We focus on using transformers to approximate both the classical CKY algorithm and this simpler variant.

## 3   Predicting Parses

As described in Section 2, we define $\beta^* \in \{0,1\}^{T \times T \times |\mathcal{N}|}$ to be a chart-sized tensor representing a CKY parse of a sentence $x$, with $\beta^*[i,j,a] = 1$ if and only if nonterminal $a$ yields $x_{i:j}$. We will use a very simple approach to predict $\beta^*$ with a transformer.

Let $\mathbf{h}_i \in \mathbb{R}^d$ be an encoder-only transformer's final-layer representation of the $i$-th token in $x$, and let $\mathbf{h}_{i,:d/2}$ and $\mathbf{h}_{i,d/2:}$, both in $\mathbb{R}^{d/2}$, be (respectively) the first and last $d/2$ elements in $\mathbf{h}_i$. We

then define $\hat{\beta}_{ij} \in \Delta^{|\mathcal{N}|}$ as

$$\hat{\beta}_{ij} \overset{\text{def}}{=} \text{softmax}(\text{FFN}([\mathbf{h}_{i,:d/2}; \mathbf{h}_{j,d/2:}])), \quad (1)$$

where above we have concatenated the first half of $\mathbf{h}_i$ and the second half of $\mathbf{h}_j$, and where FFN is a BERT-like (Devlin et al., 2019) classification head comprising a single hidden-layer, GELU non-linearity (Hendrycks and Gimpel, 2016), Layer-Norm (Ba et al., 2016), and final linear projection to $|\mathcal{N}|+1$ scores. This classification head produces a score for each nonterminal label as well as for a *non-constituent* label. This representation is similar to (but distinct from) that computed by Kitaev and Klein (2018) before using CKY on top of the computed scores. We set the lower triangle of $\hat{\beta}$ (i.e., along the first two dimensions) to zero, and use it as our approximation of $\beta^*$.

**Decoding**   Note that forming $\hat{\beta}$ from the $\mathbf{h}_i$ is only quadratic in sentence-length, and so if we can extract a parse from it in sub-cubic time we will improve (asymptotically) over CKY. In practice, we simply take the highest-scoring label for each span, and ignore spans for which the highest-scoring label is *non-constituent*. We then sort the predicted spans, first in decreasing order of end-token and then (stably) in increasing order of start-token, and build up the tree from left to right. We thus incur only an $O(T^2(|\mathcal{N}| + \log_2 T))$ decoding cost.[1]

**Training**   Given a gold (binarized) parse $\beta^*$, we simply treat predicting the labels of each span as $T(T + 1)/2$ independent multi-class classifi-

---

[1]Note this procedure is only necessary for constituency parsing, where unary productions are allowed. For parsing PCFGs in CNF we simply take the $T-1$ highest scoring spans.

cation problems, and we use the standard cross-entropy loss. Note that Corro (2020) proposes this independent-span-classification training approach in the context of discontinuous constituency parsing (though with a fixed zero-weight for the *non-constituent* label). We note that concurrent work by Yang and Tu (2023) also explores both training and decoding by making span predictions independently.

## 3.1 Decoding Parses from Chart Gradients

As described in Section 2 (also see Appendix A), a CKY parse is a subgradient of the "max score" partition function, which calculates the maximum log (joint) probability under a PCFG of a sentence and its parse tree. If we are interested in making CKY-like predictions, then, it may be a useful inductive bias to form a predicted parse from the *gradient* of some scoring function, just as CKY does. We propose to use a transformer to define this scoring function, and to predict a parse from the gradients of this scoring function with respect to its inputs.

More concretely, again letting $\mathbf{h}_i$ be an encoder-only transformer's final-layer representation of the $i$-th token in sentence $x$, we define the following "inner" score of $x$:

$$\text{score}(x) \stackrel{\text{def}}{=} \text{FFN}(1/T \sum_{i=1}^{T} \mathbf{h}_i),$$

where FFN is a BERT-like classification head producing only a single logit. Thus, we simply mean-pool over the transformer's final-layer token representations, and feed them to a feed-forward network to obtain a scalar score.

Let $\mathbf{h}_i^{(l)}$ denote an encoder-only transformer's $l$-th layer representation of the $i$-th token. Since score is a differentiable function of all the $\mathbf{h}_i^{(l)}$, we may take gradients with respect to them. In particular, let $\mathbf{g}_i \stackrel{\text{def}}{=} \frac{1}{L} \sum_{l=1}^{L} \nabla_{\mathbf{h}_i^{(l)}} \text{score}(x)$. That is, $\mathbf{g}_i$ is the average of the gradients of score with respect to each transformer layer's representation of the $i$-th token. We can then form span-predictions from the $\mathbf{g}_i$, substituting them for the $\mathbf{h}_i$ in Equation (1), to obtain

$$\hat{\beta}^+{}_{ij} \stackrel{\text{def}}{=} \text{softmax}(\text{FFN}([\mathbf{g}_{i,:d/2}; \mathbf{g}_{j,d/2:}])). \quad (2)$$

In the remainder of the paper, we refer to models making use of Equation (2) as "grad decoding."

Training a grad decoding model requires back-propagating parameter gradients through a back-propagation with respect to the $\mathbf{h}_i^{(l)}$. Fortunately, this is now simple to achieve with modern auto-differentiation frameworks, such as Pytorch (Paszke et al., 2019), which we use in all experiments.

**Discussion** It is worth noting that while $\beta^*$ is itself a subgradient of the max score partition function, our proposal above merely *decodes* $\hat{\beta}^+$ from the gradient of the inner score function. That is, the gradients $\mathbf{g}_i$ are concatenated and fed into an additional classification head, which is used to produce something chart-sized. The reason for this discrepancy is computational: if we were to feed something chart-sized into our score function, and if score required running a transformer over its input, our approach would be quartic in $T$. Instead, our approach retains quadratic dependence on $T$. Because it involves calculating gradients with respect to the $LTd$-sized transformer representations, however, it is in practice more expensive than using Equation (1); see Appendix C for details.

## 4 Constituency Parsing Experiments

We conduct experiments in two main settings. We first consider modern neural constituency parsing on standard benchmark datasets. We then consider parsing under randomly generated PCFGs. We highlight several important differences between these two settings. First, modern constituency parsing is grammarless. As such, modern constituency parsers do not predict the highest scoring parse under some PCFG, and they use a simpler variant of CKY which composes spans but has no notion of grammar-rules. While it is still interesting (at least from a computational efficiency perspective) to see if directly predicting parses is competitive with running this simpler CKY variant, it is difficult to distinguish a transformer learning this CKY variant from it simply learning to predict gold parses. This concern motivates our second setting, of randomly generated PCFGs, where there are well-defined highest-scoring parses under each PCFG, and where we can evaluate whether the transformer has predicted them. We consider this random PCFG setting in Section 5.

**Datasets** We conduct constituency parsing experiments on the English Penn Treebank (PTB; Marcus et al., 1993), the Chinese Penn Treebank (CTB; Xue et al., 2005), as well as the treebanks in the SPMRL 2013 and 2014 shared tasks (Seddah et al.,

| | English | | Chinese | | German | | Korean | |
|---|---|---|---|---|---|---|---|---|
| | Dev Set | Test Set | Dev Set | Test Set | Dev Set | Test Set | Dev Set | Test Set |
| Kitaev et al. (2019)[a] | - | 95.59 | - | 91.75 | - | 90.20 | - | 88.80 |
| Kitaev et al. (2019)[b] | **95.61** | 95.48 | **94.23** | **92.13** | 93.39 | **90.32** | **89.74** | 88.55 |
| Ours | 95.50 | **95.64** | 93.17 | 90.54 | **93.47** | 90.13 | **89.74** | **89.05** |
| Ours + grad decode | 95.07 | 95.16 | 93.75 | 91.25 | 93 | 89.63 | 89.26 | 88.46 |

Table 1: Comparison of $F_1$ score on PTB, CTB, and the German and Korean treebanks from the SPMRL 2014 shared task. All models use the same pretrained initialization; see text for details. The Kitaev et al. (2019)[a] results are those reported in the paper, which make use of an additional factored self-attention layer, while Kitaev et al. (2019)[b] are the results of running their code without this additional layer.

2013). We use the standard dataset splits throughout.

**Model Details** We adapt the chart parser first proposed by Kitaev and Klein (2018), and later refined by Kitaev et al. (2019) to involve fine-tuning a pretrained model. In particular, while Kitaev et al. (2019) fine-tune a pretrained BERT model (Devlin et al., 2019) using a margin-based structured loss between the CKY parse and the gold parse, we instead fine-tune BERT to simply predict the parse as in Equation (1), and we train with the independent-span cross-entropy loss described in Section 3. Some of the experiments in Kitaev et al. (2019) also make use of an additional factored self-attention layer that consumes the BERT representations, but we do not use this layer when predicting according to Equations (1) or (2).

Our implementation is a modification of the public Kitaev et al. (2019) implementation,[2] which also forms our main baseline. While this parser is not always state-of-the-art, it is quite close, and state-of-the-art parsers generally make use of its architecture and approach (Mrini et al., 2020; Tian et al., 2020).

The pretrained models used to initialize both the Kitaev et al. (2019) model and our own for the English and Chinese treebanks are BERT-large-uncased (Devlin et al., 2019) and BERT-base-chinese[3], respectively; BERT-base-multilingual-cased (Devlin et al., 2019) is used to initialize models for Korean, German, and the rest of the SPMRL treebanks (see Appendix D). As in the implementation of Kitaev et al. (2019), we train with AdamW (Loshchilov and Hutter, 2018; Kingma and Ba, 2015) until validation parsing performance stops increasing. We use the same learning rate scheduler as suggested in Kitaev et al. (2019),

[2]https://github.com/nikitakit/self-attentive-parser

[3]https://huggingface.co/bert-base-chinese

| Inference Speed (sent/s) | PTB | CTB |
|---|---|---|
| Kitaev et al. (2019) | 485 (1.00x) | 704 (1.00x) |
| Ours | 949 (1.96x) | 1466 (2.08x) |

Table 2: Inference speed (in sentences/second) of the CKY-based Kitaev et al. (2019) parser and our own, averaged over the PTB and CTB development sets.

which starts with 160 steps of warm-up, then decreases the learning rate by multiplying it by 0.5 when the $F_1$ score stops improving. We used a grid-search to select hyperparameters, and we provide the grid search details and the optimal hyperparameters found in Appendix F.

**Results** In Table 1 we report parsing results on these datasets, using the standard evalb evaluation. We find that our approach is competitive with the Kitaev et al. (2019) approach, slightly outperforming it on the English and Korean datasets, and slightly underperforming it on the Chinese and German datasets (see Appendix D for the results on the rest of the SPMRL treebanks). In this setting, grad decoding does not improve over simply using Equation (1). However, as we discuss in Section 5, sharing transformer layers is important in seeing the benefits of grad decoding, which we cannot do effectively with pretrained BERT models.

The fact that our simplest approach is competitive on these constituency parsing benchmarks is interesting, given that it is much faster. In Table 2 we compare the speed of our approach (in sentences parsed per second) to that of the baseline parser, and we see it is roughly two times faster. These numbers reflect the time necessary to parse the PTB and CTB development sets using the Kitaev et al. (2019) parser, and using our modification of it. Both experiments were run on the same machine, using an NVIDIA RTX A6000 GPU. We also emphasize that this comparison is somewhat favorable to the baseline parser, since we use the original

| | PTB | CTB |
|---|---|---|
| Kitaev et al. (2019) | 89.83 | 81.72 |
| Ours | **90.07** | 80.98 |
| Ours + SL + grad decode | 89.91 | **81.76** |

Table 3: Performance of our approaches and baseline when trained from scratch and evaluated in terms of $F_1$ on the PTB and CTB development sets (respectively). "SL" indicated that transformer layers are shared.

code's batching, whereas our approach makes it extremely easy to create big, padded batches, and thus speed up parsing further. Additionally, we made a comparison with Supar's (Zhang et al., 2020b,a) implementation, which we did not include since it was slower than Kitaev et al. (2019).

While the above experiments all fine-tune pretrained BERT-style models for parsing, it is also worth examining the performance of these models when trained from scratch. We accordingly train transformers from scratch using models of the same size as those in Table 1 on the PTB and CTB training sets, and report results on the development sets in Table 3. We find again that non-CKY based models are competitive. Furthermore, since we can now easily share transformer layers without sacrificing the advantage of pretrained layers, grad decoding has a more positive effect. We did not see a corresponding benefit to sharing transformer layers when predicting *without* grad decoding.

## 5 Random PCFG Experiments

To generate random PCFGs, we follow the method used by Clark and Fijalkow (2021).[4] Their method involves first generating a synthetic context-free grammar (CFG) with a specified number of terminals, non-terminals, binary rules, and lexical rules. To assign probabilities to the rules, they then use an EM-based estimation procedure (Lari and Young, 1990; Carroll and Charniak, 1992) to update the production rules such that the length distribution of the estimated PCFG is similar to that of the PTB corpus (Marcus et al., 1993).

### 5.1 Data

The approach of Clark and Fijalkow (2021) allows us to construct random grammars with a desired number of nonterminals and rules, and we generate grammars having 20 nonterminals,[5] and 100, 400,

---

[4]See https://github.com/alexc17/syntheticpcfg.

[5]We found performance patterns were the same when using more than 20 nonterminals, though results are slightly higher

and 800 rules, respectively. The number of terminals and lexical rules is set to 5000. Having more rules per nonterminal generally increases ambiguity, and so we would expect a synthetic grammar with 800 rules to be significantly more difficult to parse than one with 100, and a synthetic grammar with 20 nonterminals to be more difficult than one with more.

It is common to quantify ambiguity in terms of the conditional entropy of parse trees given sentences (Clark and Fijalkow, 2021), which can be estimated by sampling trees from the PCFG and averaging the negative log conditional probabilities of the trees given their sentences. The conditional entropy of a PCFG $G$ is thus estimated as

$$\hat{H}_G(\tau|x) = -\frac{1}{N} \sum_{n=1}^{N} \log \frac{p_G(x^{(n)}, \tau^{(n)})}{p_G(x^{(n)})},$$

where the $(x^{(n)}, \tau^{(n)})$ are sampled from $G$, and where $p_G(x^{(n)})$ is calculated with the inside algorithm. We use $N = 1000$ samples. Below we report the $\hat{H}_G(\tau|x)$ of each random grammar along with parsing performance; we will see that $\hat{H}_G(\tau|x)$ negatively correlates with performance.

Our datasets consist of sentences sampled from these PCFGs and their corresponding parses, which were parsed with the CKY algorithm.

**Dataset size** Because we are interested in testing the ability of transformers to capture the CKY algorithm, we must ensure that the training set is sufficiently large that prediction errors can be attributed to the transformer failing to learn the algorithm, and not to sparsity in the training data. We ensure this by simply adding data until validation performance plateaus. In particular, we use ~200K sentences for training and 2K held-out sentences. To control the complexity of our datasets, we also limit the maximum sentence length to 30, which decreases the average sentence length from 20-25 words per sentence to 18.

### 5.2 Models and baselines

Whereas the experiments in the previous section mostly make use of standard BERT-like architectures, either fine-tuned or trained from scratch, in this section we additionally consider making parse predictions (as in Equations (1) and (2)) with transformer variants which are intended to improve performance on "algorithmic" tasks. Indeed, recent

---

since the grammars become *less* ambiguous.

| $\vert\mathcal{R}\vert$ | 100 | | 400 | | 800 | |
|---|---|---|---|---|---|---|
| $\hat{H}_G(\tau\vert x)$ | 0.35 | | 3.99 | | 9.74 | |
| | $F_1$ | $\Delta_{\mathrm{lp}}$ | $F_1$ | $\Delta_{\mathrm{lp}}$ | $F_1$ | $\Delta_{\mathrm{lp}}$ |
| CKY | 100.0 | 0 | 100.0 | 0 | 100.0 | 0 |
| Transformer | 98.31 | 0.04 | 87.86 | 0.29 | 74.80 | 0.45 |
| Transformer + SL | 98.61 | 0.03 | 89.89 | 0.19 | 79.91 | 0.29 |
| Transformer + SL + CG | 98.97 | 0.02 | 89.43 | 0.21 | 78.64 | 0.31 |
| Transformer + SL + GD | **99.06** | **0.01** | **91.07** | **0.17** | **81.78** | **0.28** |

Table 4: Labeled span $F_1$ performance (as in `evalb`) on randomly generated PCFGs with $\vert\mathcal{N}\vert = 20$ nonterminals, and $\vert\mathcal{R}\vert = 100$, 400, and 800 rules, respectively. $\hat{H}_G(\tau\vert x)$ indicates the estimated conditional entropy of the grammar. $\Delta_{\mathrm{lp}}$ is the average log probability difference between CKY and predicted parses. "SL" indicates that transformer layers are shared, "CG" that a copy-gate is used, and "GD" that grad decoding is used.

research has shown that transformers often fail to generalize on algorithmic tasks (Saxton et al., 2019; Hupkes et al., 2020; Dubois et al., 2020; Chaabouni et al., 2021), which has motivated architectural modifications, such as the Universal Transformer (UT; Dehghani et al., 2018) and the Neural Data Router (NDR; Csordás et al., 2021).

One major architectural modification common to both UT and NDR is that all transformer layers share the same parameters; this modification is intended to capture the intuition that recursive computation often requires applying the same function multiple times. NDR additionally makes use of a "copy gate," which allows transformer representations at layer $l$ to be simply copied over as the representation at layer $l + 1$ without being further processed, as well as "geometric attention," which biases the self-attention to attend to nearby tokens. We found geometric attention to significantly hurt performance in preliminary experiments, and so we report results only with the shared-layer and copy-gate modifications.

All models in this section are trained from scratch. This is convenient for UT- and NDR-style architectures, for which we do not have large pre-trained models, but we also found in preliminary experiments that on random PCFGs pretrained vanilla transformers (such as BERT) did not improve over transformers trained from scratch. We did find a modest benefit to fine-tuning a BERT-style model that we pretrained on sentences from our randomly generated grammars (rather than on natural language), which accords with the recent work of Zhao et al. (2023); see Appendix E for details. Our implementation is based on the BERT implementation in the Hugging Face `transformers` library (Wolf et al., 2020).

| | sentences/second | | |
|---|---|---|---|
| batch size | 32 | 64 | 128 |
| CKY | 698 | 1162 | 1576 |
| Ours | 1817 | 2132 | 2161 |

Table 5: Inference speed of GPU-based `torch-struct` CKY parsing and our transformer-based approach on a random grammar with $\vert\mathcal{R}\vert = 400$ rules. We report the median speed of running our model and CKY 5 times on 60K samples in batches of 32, 64, and 128, respectively.

**Evaluation** We evaluate our predicted parses against the CKY parse returned by `torch-struct` (Rush, 2020), using $F_1$ over span-labels as in `evalb`. Since there may be multiple highest-scoring parses, we also report the average difference in log probability, averaged over each production, between the CKY parse returned by `torch-struct` and the predicted parse. We refer to this metric as "$\Delta_{\mathrm{lp}}$" in tables. Because transformer-based parsers may predict invalid productions, we smooth the PCFG by adding $10^{-5}$ to each production probability and renormalizing before calculating $\Delta_{\mathrm{lp}}$.

**Results** The results of these synthetic parsing experiments are in Table 4, where we show parsing performance over three random grammars with different conditional entropies. We see that transformers struggle as the ambiguity increases, suggesting that these models are not in fact implementing CKY-like processing. At the same time, we see that incorporating inductive bias does help in nearly all cases, and that gradient decoding together with sharing transformer layers performs best for all grammars.

We conduct a speed evaluation in Table 5, where we compare with the CKY implementation in `torch-struct` (Rush, 2020), which is optimized

for performance on GPUs. For a fixed number of nonterminals, neither the speed of `torch-struct`'s CKY implementation nor of our transformer-based approximation is significantly affected by the number of rules. Accordingly, we show speed results for parsing just the synthetic grammar with 400 rules, for three different batch-sizes. All timing experiments are run on the same machine and utilize an NVIDIA RTX A6000 GPU. We see that the transformer-based approximation is always faster, although its advantage decreases as batch-size increases. We note, however, that because our implementation uses Hugging Face `transformers` components (Wolf et al., 2020), which at present do not make use of optimization such as FlashAttention (Dao et al., 2022), our approach could likely be sped up further.

## 6 Related Work

Modern neural constituency parsers typically fall into one of three camps: chart-based parsers (Stern et al., 2017; Gaddy et al., 2018; Kitaev and Klein, 2018; Mrini et al., 2020; Tian et al., 2020; Tenney et al., 2019; Jawahar et al., 2019; Li et al., 2020; Murty et al., 2022), transition-based parsers (Zhang and Clark, 2009; Cross and Huang, 2016; Vaswani and Sagae, 2016; Vilares and Gómez-Rodríguez, 2018; Fernandez Astudillo et al., 2020), or sequence-to-sequence-style parsers (Vinyals et al., 2015; Kamigaito et al., 2017; Suzuki et al., 2018; Fernández-González and Gómez-Rodríguez, 2020; Nguyen et al., 2021; Yang and Tu, 2022).

Our approach most closely resembles chart-based methods, in that we compute scores for all spans for all non-terminals. However, unlike chart-based parsers, we aim to only approximate, or amortize, running the CKY algorithm. Unlike transition-based or sequence-to-sequence-style methods, our approximation involves attempting to predict a parse jointly and independently, rather than incrementally. Within the world of chart-based neural parsers, those of Kitaev and Klein (2018) and Kitaev et al. (2019) have been enormously influential, and state-of-the-art constituency parsers, such as those of Mrini et al. (2020) and Tian et al. (2020), adapt this approach while improving it.

The approach of employing an independent-span classification training objective along with greedy decoding for inference has also been explored by Zhang et al. (2019) in the context of dependency parsing. It is worth emphasizing that while the Zhang et al. (2019) work shows that a scoring model with quadratic complexity can approximate the also quadratic Chu-Liu-Edmonds algorithm (Chu and Liu, 1965), used for decoding in dependency parsing, our work focuses on approximating a cubic complexity decoding algorithm using a scoring model with quadratic complexity.

We are additionally motivated by recent work on neural algorithmic reasoning (Xu et al., 2019; Csordás et al., 2021; Dudzik and Veličković, 2022; Delétang et al., 2022; Ibarz et al., 2022; Liu et al., 2023), some of which has endeavored to solve classical dynamic programs (Veličković et al., 2022) and MDPs (Chen et al., 2021) with graph neural networks and transformers, respectively. One respect in which learning to compute CKY differs from many other algorithmic reasoning challenges (including other dynamic programs) is that in addition to its discrete sentence input, CKY also consumes continuous log potentials.

Finally, the idea of training a model to produce the solution to an optimization problem is known as training an "inference network," and has been used famously in approximate probabilistic inference scenarios (Kingma and Welling, 2014) and in approximating gradient descent (Johnson et al., 2016). Most similar to our approach, Tu and Gimpel (2018) train an inference network to do Viterbi-style sequence labeling, although they do not consider parsing, and they require inference networks for both training and test-time prediction due to their large-margin approach.

## 7 Discussion and Conclusion

Our findings on the ability of transformers to approximate CKY are decidedly mixed. On the one hand, using transformers to independently predict spans in a constituency parse is competitive with using very strong neural chart parsers, and it is moreover much faster to predict parses in this independent, non-CKY-based way. On the other hand, if transformers are capturing the CKY computation, they ought to be able to parse even under random PCFGs, and it is clear that as the ambiguity of the grammar increases they struggle with this.

We have also found that making a transformer's computation more closely resemble that of a classical algorithm, either by sharing computation layers as proposed by Dehghani et al. (2018) and Csordás et al. (2021), or by having it make use of the

gradient with respect to a scoring function, is helpful. This finding both confirms previous results in this area, and also suggests that the inductive bias we seek to incorporate in our models may need to closely match the problem.

There are many avenues for future work, and attempting to find a minimal general-purpose architecture that *can* in fact parse under random PCFGs is an important challenge. In particular, it is worth exploring whether other forms of pretraining (i.e., pre-training distinct from BERT's) might benefit this task more. Another important future challenge to address is whether it is possible to have a model consume both the input sentence as well as the parameters (as the CKY algorithm does), rather than merely pretrain on parsed sentences generated using the parameters.

## Limitations

A limitation of the general paradigm of learning to compute algorithmically is that it requires a training phase, which can be expensive computationally, and which requires annotated data. This is less of a limitation in the case of constituency parsing, however, since we are likely to be training models in any case.

Another important limitation of our work is that we have only provided evidence that transformers are unable to implement CKY in our particular experimental setting. While we have endeavored to find the best-performing combinations of models and losses (and while this combination appears to perform well for constituency parsing), it is possible that other transformer-based architectures or other losses could significantly improve in terms of parsing random grammars.

We also note that a limitation of the grad decoding approach we propose is that we have found that it is more sensitive to optimization hyperparameters than are the baseline approaches.

Finally, we note that our best-performing constituency results make use of large pretrained models. These models are expensive to train, and do not necessarily exist for all languages we would like to parse.

## Ethics Statement

We do not believe there are significant ethical issues associated with this research, apart from those that relate to training moderately-sized machine learning models on GPUs.

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

## A   Additional background on CKY

Letting $\ell^R \in \mathbb{R}^{|\mathcal{R}| \times |\mathcal{N}| \times |\mathcal{N}|}$ represent the log potentials (e.g., log probabilities) associated with the rules in PCFG $G$, and $\ell^E(x) \in \mathbb{R}^{T \times |\mathcal{N}|}$ the log potentials corresponding to each token in input sentence $x$, we compute the chart $\beta \in \mathbb{R}^{T \times T \times |\mathcal{N}|}$ for $x$, where $\beta[i, j, a]$ represents the sum (under a particular semiring) of all weight associated with the $a$-th non-terminal yielding $x_{i:j}$. In particular, with semiring operations $\oplus$ and $\otimes$, and if $1 \leq i < j \leq T$, we have $\beta[i, j, a] = \bigoplus_{k,b,c} \ell^R_{a,b,c} \otimes \beta[i, k, b] \otimes \beta[k + 1, j, c]$. Assuming the first slice of $\ell^R$ along the first dimension (i.e., $\ell^R_{1,:,:}$) corresponds to rules with $S$ on the left-hand-side, the log partition function is then given by $A(\ell^R, \ell^E) = \beta[1, T, 1]$.

Furthermore, under the max-plus semiring, one of the subgradients of $A(\ell^R, \ell^E)$ with respect to $\beta$ is a one-hot representation of a highest scoring parse for $x$ under $G$ (Eisner, 2016; Rush, 2020). We refer to such a one-hot subgradient as $\beta^* \in \{0, 1\}^{T \times T \times |\mathcal{N}|}$. Rush (2020) therefore proposes to compute $\beta^*$ using automatic differentiation, and provides a fast implementation tuned for use on GPUs.

## B   Model, Training, and Dataset Details

**Dataset Details**   We provide details on the standard constituency parsing datasets used in our experiments in Table 6.

|        | train  | dev   | test  |
|--------|--------|-------|-------|
| PTB    | 39,831 | 1,700 | 2,416 |
| CTB    | 17,544 | 352   | 348   |
| German | 40,472 | 5,000 | 5,000 |
| Korean | 23,010 | 2,066 | 2,287 |

Table 6: Number of examples in the standard splits of the English Penn Treebank (Marcus et al., 1993), the Chinese Penn Treebank (Xue et al., 2005), and the SPMRL Treebanks (Seddah et al., 2013).

**Terms of Use**   We used the standard English Penn Treebank, Chinese Penn Treebank, and treebanks (except for Arabic) from SPMRL 2013 and 2014 shared tasks in accordance with their licenses. Both PTB and CTB are under the Linguistic Data Consortium (LDC) licenses. The German, Hebrew, Korean, and Swedish Treebank are not under any specific licenses. The Basque Treebank is licensed under the Creative Commons license. The Polish Treebank is licensed under GPL v3. We also use Hugging Face code and models in accordance with

their licenses (Apache 2.0). For our deep learning framework, we use PyTorch (Paszke et al., 2019) which is under the BSD-3 license. We also make use of code provided by Kitaev et al. (2019)[6]. Their code is available under the MIT license.

**Computational Budget**   All of our experiments were run on NVIDIA RTX A6000 GPUs. We provide the computational time of our experiments on constituency parsing datasets and a random PCFG. Since the training times of random PCFGs with $|\mathcal{R}| = 100$, 400, and 800 rules are similar, we only provide that of the PCFG with $|\mathcal{R}| = 400$.

**Models Details**   In Table 7 and 8, we provide the number of model parameters used in constituency parsing and random PCFG experiments, respectively.

|       | Model                        | Size |
|-------|------------------------------|------|
| PTB   | BERT-large-uncased           | 345M |
| CTB   | BERT-base-chinese            | 102M |
| SPMRL | BERT-base-multilingual-cased | 178M |

Table 7: Model sizes for constituency parsing experiments. All models are initialized from the Hugging Face `transformers` library (Wolf et al., 2020).

| Model                    | Size  |
|--------------------------|-------|
| Transformer              | 89.6M |
| Transformer + SL         | 11.1M |
| Transformer + SL + CG    | 11.9M |
| Transformer + SL + GD    | 11.7M |

Table 8: Model sizes for random PCFG experiments.

|                   | PTB | CTB | German | Korean |
|-------------------|-----|-----|--------|--------|
| Ours              | 1   | 0.5 | 4.5    | 1.5    |
| Ours + grad decode| 8   | 4.5 | 8.5    | 2.5    |

Table 9: Approximate total training time (in number of hours) of our model with and without grad decoding on the parsing datasets. We find that grad decoding models take longer to converge.

## C   Gradient Decoding Details

**Speed Comparison**   In Table 11 we compare the inference time between our models (with and without gradient decoding) and CKY.

---

[6] https://github.com/nikitakit/self-attentive-parser

| | $|\mathcal{R}| = 400$ |
|---|---|
| Transformer | 11 |
| Transformer + SL | 7 |
| Transformer + SL + CG | 6 |
| Transformer + SL + GD | 15 |

Table 10: Approximate total training time (in number of hours) of our model on a random PCFG. "SL" indicates that transformer layers are shared, "CG" that a copy-gate is used, and "GD" that grad decoding is used.

| | sentences/second | | |
|---|---|---|---|
| batch size | 32 | 64 | 128 |
| CKY | 698 | 1162 | 1576 |
| Ours | 1817 | 2132 | 2161 |
| Ours + GD | 1111 | 1524 | 1632 |

Table 11: Inference speed of GPU-based `torch-struct` CKY parsing and our transformer-based approach on a random grammar with $|\mathcal{R}| = 400$ rules. We report the median speed of running our models and CKY 5 times on 60K samples in batches of 32, 64, and 128, respectively.

**Memory Comparison** The maximum memory allocation of our model with gradient decoding is 1.52 GB, which is only $10\%$ higher compared to our model without gradient decoding, which has a maximum memory allocation of 1.38 GB.

## D  Constituency Parsing Results on SPMRL

Table 12 shows that our approach outperforms Kitaev et al. (2019) on half of the SPMRL treebanks. We excluded the Arabic treebank since we were unable to get its corresponding license.

Considering that gradient decoding did not lead to substantial improvements in the results for PTB, CTB, German, and Korean, as indicated in Table 1, we decided not to perform the gradient decoding experiments on the rest of the SPMRL treebanks due to limitations in computational resources.

## E  MLM Pretraining on Random PCFG-Generated Data

Several works have suggested that pretrained models can capture syntactic information (Hewitt and Manning, 2019; Manning et al., 2020; Maudslay and Cotterell, 2021; Zhao et al., 2023). In particular, Zhao et al. (2023) argued that a connection exists between MLM and the inside-outside algorithm. Through probing, they show that models

pretrained on synthetic PCFG data may be approximating the inside-outside algorithm. Since inside-outside and CKY are related, it is natural to question whether MLM pretraining can also be helpful when it comes to approximating CKY.

We therefore pretrain a transformer model on 500K sentences generated from the random PCFG with 800 rules. We selected our training setup following the Cramming training recipe (Geiping and Goldstein, 2022). For training, we utilized a large batch size of 4096, accumulating gradients and performing an update every 32 steps. We employed a linear warmup schedule for $10\%$ of the total training steps, with a peak learning rate of $1 \times 10^{-4}$. The model achieved a training perplexity of 92.01 and a validation perplexity of 92.20.

In Table 13, we show that fine-tuning the pretrained model leads to performance improvements, particularly when layers are not shared. It is important to highlight that while pretraining is helpful, it comes with higher computational costs compared to relying solely on inductive biases, and it demonstrates less improvement compared to gradient decoding.

## F  Hyperparameters

Table 14 shows the grid we used to search for the optimal hyperparameters of the models used in constituency parsing and random PCFG experiments. We also provide the optimal hyperparameters in Table 15 and 16. For the baseline result (Kitaev and Klein, 2018) in Table 1, we used the hyperparameters recommended in their code. The results in Table 1 and 4 make use of the optimal hyperparameters and are based on a single run using a fixed seed throughout all experiments.

|  | German | Korean | Basque | French | Hebrew | Hungarian | Polish | Swedish |
|---|---|---|---|---|---|---|---|---|
| Kitaev et al. (2019) | **90.20** | 88.80 | 90.70 | 87.35 | **92.95** | 94.60 | **96.26** | **89.94** |
| Ours | 90.13 | **89.05** | **90.93** | **87.59** | 92.69 | **94.64** | 95.86 | 89.29 |

Table 12: Comparison of $F_1$ score on the test sets of the SPMRL treebanks. The Kitaev et al. (2019) results are those reported in the paper.

|  | $|\mathcal{R}| = 800$ |
|---|---|
| Transformer | 78.14 |
| Transformer + SL | 78.88 |
| Transformer + SL + GD | 80.64 |

Table 13: $F_1$ performance of pretrained models on randomly generated PCFGs with $|\mathcal{N}| = 20$ nonterminals, and $|\mathcal{R}| = 800$ rules.

| Ours | | | | | |
|---|---|---|---|---|---|
|  | scheduler | LR | grad clip | WD | AD |
| English | linear | 5e-5 | 0.3 | 0 | 0.1 |
| Chinese | default | 6e-5 | 0.4 | 0 | 0.1 |
| German | default | 5e-5 | 0.3 | 0 | 0.1 |
| Korean | default | 5e-5 | 0 | 0 | 0.1 |
| Basque | default | 5e-5 | 0.3 | 0 | 0.1 |
| French | default | 6e-5 | 0 | 0 | 0.1 |
| Hebrew | default | 5e-5 | 0.5 | 0 | 0.1 |
| Hungarian | default | 5e-5 | 0.3 | 0 | 0.1 |
| Polish | default | 5e-5 | 0.3 | 0 | 0.1 |
| Swedish | default | 6e-5 | 0.3 | 0 | 0.1 |
| Ours + grad decode | | | | | |
| English | default | 7e-5 | 0.2 | 0 | 0.1 |
| Chinese | default | 8e-5 | 0.3 | 0 | 0.1 |
| German | default | 8e-5 | 0.3 | 0 | 0.1 |
| Korean | default | 7e-5 | 0.2 | 0 | 0.1 |
| Kitaev et al. (2019) | | | | | |
| All | default | 5e-5 | 0 | 0.001 | 0.2 |

Table 15: Optimal hyperparameters used for our constituency parsing experiments. We denote the learning rate as "LR", weight decay as "WD", and attention dropout as "AD". Similar hyperparameters are used in the baseline model (Kitaev et al., 2019) across the datasets. Results are provided in Table 1.

| Constituency parsing Experiments | |
|---|---|
| Hyperparameter | Values |
| Scheduler | [default[a], linear[b]] |
| Learning rate | [5e-5, 6e-5, 7e-5, 8e-5] |
| Gradient clipping | [0, 0.2, 0.3, 0.4] |
| Weight decay | [0, 1e-3, 1e-2] |
| Attention dropout | [0.1, 0.2] |
| Random PCFG Experiments | |
| Scheduler | constant + warmup |
| Learning rate | [1e-4, 1.3e-4, 1.5e-4, 1.7e-4] |
| Gradient clipping | [0.3, 0.4, 1] |
| Weight decay | [0, 1e-3, 1e-2] |
| Attention dropout | [0.1, 0.2] |
| Warmup steps | [2000, 4000] |

Table 14: The grid we used to search for the optimal hyperparameters in our constituency parsing and random PCFG experiments. [a]The default scheduler used in Kitaev et al. (2019) as mentioned in section 4. [b]160 steps of warm-up then decreasing the learning rate linearly.

| Transformer | | | | | |
|---|---|---|---|---|---|
|  | LR | grad clip | WD | AD | WS |
| $|\mathcal{R}| = 100$ | 1e-4 | 1 | 1e-3 | 0.3 | 4000 |
| $|\mathcal{R}| = 400$ | 1e-4 | 1 | 1e-3 | 0.3 | 4000 |
| $|\mathcal{R}| = 800$ | 1e-4 | 1 | 1e-3 | 0.3 | 4000 |
| Transformer + SL | | | | | |
| $|\mathcal{R}| = 100$ | 1e-4 | 1 | 1e-3 | 0.3 | 4000 |
| $|\mathcal{R}| = 400$ | 1.3e-4 | 1 | 1e-3 | 0.3 | 4000 |
| $|\mathcal{R}| = 800$ | 1.3e-4 | 1 | 1e-3 | 0.3 | 4000 |
| Transformer + SL + CG | | | | | |
| $|\mathcal{R}| = 100$ | 1.3e-4 | 1 | 1e-3 | 0.3 | 4000 |
| $|\mathcal{R}| = 400$ | 1e-4 | 1 | 1e-3 | 0.3 | 4000 |
| $|\mathcal{R}| = 800$ | 1e-4 | 1 | 1e-3 | 0.3 | 4000 |
| Transformer + SL + GD | | | | | |
| $|\mathcal{R}| = 100$ | 1e-4 | 0.4 | 1e-3 | 0.3 | 4000 |
| $|\mathcal{R}| = 400$ | 1.3e-4 | 0.4 | 1e-3 | 0.3 | 4000 |
| $|\mathcal{R}| = 800$ | 1e-4 | 0.4 | 1e-3 | 0.3 | 4000 |

Table 16: Optimal hyperparameters used for our random PCFG experiments. We denote the learning rate as "LR", weight decay as "WD", attention dropout as "AD", and the number of warmup steps as "WS". Results are provided in Table 4.

