# OpenReview forum: "Approximating CKY with Transformers"
_EMNLP/2023/Conference — EMNLP 2023 Findings_

### Official Review · Reviewer_e4gq · 2023-07-28

**Soundness:** 4

**Excitement:**

3: Ambivalent: It has merits (e.g., it reports state-of-the-art results, the idea is nice), but there are key weaknesses (e.g., it describes incremental work), and it can significantly benefit from another round of revision. However, I won't object to accepting it if my co-reviewers champion it.

**Missing References:**

- In lines 76-77 and 239-247, the paper "https://aclanthology.org/C18-1011.pdf" should be cited. This work is the earliest known usage of local span classification loss for constituency parsing.

**Paper Topic And Main Contributions:**

This study investigates the feasibility of implementing CKY using Transformers. The authors conduct two experiments: (1) training a neural constituency parser and (2) training on a synthesis dataset generated by a predefined PCFG. Both experiments utilize a local span classification loss and a heuristic greedy decoding algorithm without CKY.

In the first experiment, the authors achieve competitive performance compared to span-based parsers with CKY while achieving faster parsing speed.

In the second experiment, the authors observe that Transformers perform poorly when the grammar is highly ambiguous, suggesting that Transformers cannot perfectly implement CKY. However, they propose several effective techniques, such as weight sharing across layers, copy gates, and gradient decoding, to mitigate this issue and improve the performance.

**Questions For The Authors:**

- "Physics of Language Models: Part 1, Context-Free Grammar" (https://arxiv.org/pdf/2305.13673.pdf) demonstrates that Transformers can perfectly learn Context-Free Grammar (CFG). Does this finding contradict your results?  If not, could you kindly provide a brief explanation?

- I am curious to know if "Transformers Learn Shortcuts to Automata" is related to the Transformers' ability to learn dynamic programs.

**Reasons To Accept:**

- The research topic holds significant importance as it aims to provide insights into the working mechanism of Transformers.

- The proposed gradient decoding strategy is innovative and intriguing, adding novelty and interest to the study.

**Reasons To Reject:**

- The distribution of constituency parse trees for a given sentence is highly peaked, which makes it a less suitable task to study whether Transformers can implement CKY, as acknowledged by the author in lines 81-86. The primary contribution of the first experiment in constituency parsing lies in the use of local loss and greedy decoding to improve parsing efficiency. However, it is worth noting that the parsing literature has already demonstrated that with a powerful neural encoder, parsers can achieve good performance using local classification loss and greedy decoding, as illustrated in Figure 1 of https://aclanthology.org/P19-1562.pdf and Table 5 of https://aclanthology.org/2023.acl-long.469.pdf. As a result, the significance of this particular contribution in the present work is somewhat diminished.

- My complaint about this paper is that I believe the experiments in the constituency parsing section are entirely unnecessary. Emphasizing the speed advantage does not seem appropriate as the main or even partial motivation for this article. It would be more suitable to focus on fundamental questions like whether Transformers can learn dynamic programming algorithms like CKY. However, the study of this aspect in the paper appears to be relatively superficial and not very thorough, as it does not provide me with significantly novel insights.

**Reproducibility:**

4: Could mostly reproduce the results, but there may be some variation because of sample variance or minor variations in their interpretation of the protocol or method.

**Reviewer Confidence:**

4: Quite sure. I tried to check the important points carefully. It's unlikely, though conceivable, that I missed something that should affect my ratings.

---

> ### Author Rebuttal · Authors · 2023-08-29
>
> We thank Reviewer e4gq for the thoughtful comments, questions, and suggestions.
>
> >The distribution of constituency parse trees for a given sentence is highly peaked, which makes it a less suitable task to study whether Transformers can implement CKY, as acknowledged by the author in lines 81-86. The primary contribution of the first experiment in constituency parsing lies in the use of local loss and greedy decoding to improve parsing efficiency. However, it is worth noting that the parsing literature has already demonstrated that with a powerful neural encoder, parsers can achieve good performance using local classification loss and greedy decoding, as illustrated in Figure 1 of https://aclanthology.org/P19-1562.pdf and Table 5 of https://aclanthology.org/2023.acl-long.469.pdf. As a result, the significance of this particular contribution in the present work is somewhat diminished.
>
> **Response to reason to reject #1:**
> Regarding previous demonstrations of the viability of local classification and greedy decoding: we note that our work differs from the Zhang et al. (2019) work in that we consider constituency parsing rather than dependency parsing and in that we use a transformer-based architecture rather than an LSTM-based one. As the focus of our work is precisely on the extent to which transformers approximate CKY (an algorithm not used in dependency parsing), we feel our experiments are necessary and substantially different from those of Zhang et al.
>
> Regarding the work of Yang & Tu (2023), we assume Reviewer e4gq is referring to the “greedy” ablation in Table 5, as most of the Yang & Tu paper deals with an autoregressive encoder-decoder model, unlike ours. Yang & Tu mention that this ablation is based on Teng & Zhang (2018), which Reviewer e4gq also highlights as a missing reference. We agree this is a missing reference, and appreciate Reviewer e4gq mentioning it. However, it differs from our approach in that it is a two-step parser, which first predicts an unlabeled tree (by classifying spans as constituents or not), and then predicts labels for each span using a tree-LSTM. In contrast we use a simpler single-step approach and a transformer. We also note that our approach to appears to very slightly outperform the results in Yang & Tu’s table 5, and that we include results on more languages (see response to Reviewer Fb95 for additional languages). Finally, we note that the Yang & Tu work is considered contemporaneous to ours under EMNLP’s policy (https://2023.emnlp.org/calls/main_conference_papers/), and indeed this paper was not yet published at the time we submitted.
>
> > My complaint about this paper is that I believe the experiments in the constituency parsing section are entirely unnecessary. Emphasizing the speed advantage does not seem appropriate as the main or even partial motivation for this article. It would be more suitable to focus on fundamental questions like whether Transformers can learn dynamic programming algorithms like CKY. However, the study of this aspect in the paper appears to be relatively superficial and not very thorough, as it does not provide me with significantly novel insights.
>
> **Response to reason to reject #2:**
> Regarding the necessity of the constituency parsing experiments, we believe that without these experiments many readers would wonder to what extent our results carried over to the practical constituency parsing case, and we have tried to fill out the paper by answering this question for a fairly extensive number of natural languages.
>
> Regarding superficiality and lack of thoroughness, we would be very happy to hear suggestions for further experiments from Reviewer e4gq, and we will endeavor to add them in the next revision.
>
>
> **Response to questions:**
>
> > "Physics of Language Models: Part 1, Context-Free Grammar" (https://arxiv.org/pdf/2305.13673.pdf) demonstrates that Transformers can perfectly learn Context-Free Grammar (CFG). Does this finding contradict your results? If not, could you kindly provide a brief explanation?
>
> The “Physics of Language Models” paper does not contradict our results because it considers a different setting and task. That paper considers decoder-only transformers trained as language models, and it evaluates them on the task of generating continuations of prefixes such that the resulting string belongs to a particular CFG. In contrast, we consider encoder-only transformers trained with a different loss on *weighted* CFGs (namely, on PCFGs), where the task is to parse, i.e., to predict the highest-scoring tree under the relevant PCFG. This task is not considered by the Physics of LM paper and so their results do not extend to our setting.
>
> > I am curious to know if "Transformers Learn Shortcuts to Automata" is related to the Transformers' ability to learn dynamic programs.
>
> Yes, we agree that there is an intuitive connection between the Shortcuts to Automata paper and the ability (or not) of transformers to learn dynamic programs. We agree this is an interesting avenue for future work.

---

### Official Review · Reviewer_LCCU · 2023-08-05

**Soundness:** 4

**Excitement:**

3: Ambivalent: It has merits (e.g., it reports state-of-the-art results, the idea is nice), but there are key weaknesses (e.g., it describes incremental work), and it can significantly benefit from another round of revision. However, I won't object to accepting it if my co-reviewers champion it.

**Paper Topic And Main Contributions:**

The main results of this paper use a pretrained BERT model to do supervised constituent parsing by predicting the highest-scoring constituent and ignoring non-constituents. The core chart parsing model is based on that of Kitaev and Klein (2018), however, instead of using a max-margin loss, a corss-entropy loss scoring possible constituents over spans is used. Using this approach, competitive parsing performance is shown on four languages (en, de, zh, and ko). The paper further extends the baseline model to predict random PCFG generated parses, demonstrating that by using additional structural chart constraints in the form of subgradients (Eisner, 2016) this further improves over the BERT-based baseline. The main contribution of this paper is a variation of the loss used in Kitaev and Klein (2018).

**Reasons To Accept:**

* This papers presents a model with positive results
* Positive synthetic experiments on random PCFGs

**Reasons To Reject:**

* The model is essentially a CKY parser (with subgradient constraints, Eisner, 2016) with the core being a pretrained BERT, and UT and NDR (without pretrained models) for the synthetic experiments
* All main ideas have been proposed before and used in similar ways, and the main contributions represent slight variations of them only
* The performance on random PCFGs mainly comes from UR and NDR transformers (Table 4)

**Reproducibility:**

4: Could mostly reproduce the results, but there may be some variation because of sample variance or minor variations in their interpretation of the protocol or method.

**Reviewer Confidence:**

4: Quite sure. I tried to check the important points carefully. It's unlikely, though conceivable, that I missed something that should affect my ratings.

---

> ### Author Rebuttal · Authors · 2023-08-29
>
> We thank Reviewer LCCU for the thoughtful comments.
>
> > The model is essentially a CKY parser (with subgradient constraints, Eisner, 2016) with the core being a pretrained BERT, and UT and NDR (without pretrained models) for the synthetic experiments
>
> **Response to reason to reject #1:**
> We respectfully disagree with Reviewer LCCU’s view that the model is essentially a CKY parser. Indeed, unlike the vast majority of previous work on constituency parsing, our parser never runs the CKY algorithm. As we emphasize, this has ramifications both for runtime (quadratic rather than cubic) and for understanding to what extent transformers approximate classical algorithms. Further, the idea of decoding predictions from gradients with respect to a chart representation is entirely novel, as far as we are aware.
>
> > All main ideas have been proposed before and used in similar ways, and the main contributions represent slight variations of them only
>
> **Response to reason to reject #2:**
> We emphasize that the experiments in the paper are not (only) designed to present a new architecture or loss, but to test the hypothesis that transformers are approximating the CKY algorithm. We believe our empirical findings are novel and that they bear on this important question, whether or not certain architectures or losses have also been used before in other contexts.
>
>
>
> > The performance on random PCFGs mainly comes from UR and NDR transformers (Table 4)
>
> **Response to reason to reject #3:**
> We respectfully disagree with Reviewer LCCU’s view that the performance on random PCFGs comes mainly from UT and NDR. First note that "Transformer + SL + GD" does not use "CG", which is the contribution of NDR, and that this variant improves over "Transformer + SL" by **~0.45**, **~1.18**, and **~1.87** points of F1-score, respectively, on the three synthetic grammars. We further observe that performance gap *increases* as the grammar gets harder, and as performance in general decreases.
>
> **Response to the reproducibility score:**
> We emphasize that we have provided all the hyperparameters and details on the model architectures in Appendix E and Section 4-Model Details. Furthermore, The code will be open-sourced upon publication and will include instructions for reproducing all the experiments.
> Regarding the dataset generation or any other point of confusion, we would be happy to clarify any questions or concerns that Reviewer LCCU might have in our next revision.

---

### Official Review · Reviewer_Fb95 · 2023-08-06

**Soundness:** 3

**Excitement:**

3: Ambivalent: It has merits (e.g., it reports state-of-the-art results, the idea is nice), but there are key weaknesses (e.g., it describes incremental work), and it can significantly benefit from another round of revision. However, I won't object to accepting it if my co-reviewers champion it.

**Paper Topic And Main Contributions:**

This paper investigates whether transformers can learn CKY algorithm. The authors first presented how transformer based constituency parsers work and how that information is used in CKY algorithm. Then the changes to training and inference are presented which predicts the gradient to decode the parse. Four constituent treebanks (English, Chinese, German and Korean) are considered to evaluate the hypothesis and show that their approach give competitive accuracies with double the speed of state-of-the-art parsers. Then they experiment with synthetic PCFGs and show that transformers are indeed not learning CKY internally since it fails to work for synthetic PCFGs.

**Reasons To Accept:**

This paper studies an interesting question of whether transformers approximate CKY algorithm. The experiments are thorough with detailed analysis of the results. Rather than just experimenting with English treebank, the authors experiment with four diverse treebanks which shows that the approach works for diverse languages. Similarly in addition to constituency treebanks, experiments were conducted on synthetic PCFGs which add more value to the work.

**Reasons To Reject:**

It is great to see the results on four languages. But it would be even better to have results on all the language treebanks in SPMRL 2013.

**Reproducibility:**

3: Could reproduce the results with some difficulty. The settings of parameters are underspecified or subjectively determined; the training/evaluation data are not widely available.

**Reviewer Confidence:**

3: Pretty sure, but there's a chance I missed something. Although I have a good feel for this area in general, I did not carefully check the paper's details, e.g., the math, experimental design, or novelty.

---

> ### Author Rebuttal · Authors · 2023-08-29
>
> We thank Reviewer Fb95 for the thoughtful comments and suggestions.
>
> As suggested by Reviewer Fb95, below we provide our results on all the other SPMRL datasets except for Arabic which we currently do not have access to; we will try to include it in our next revision.
> Please note that we did not do extensive hyperparameter tuning and therefore our results on these languages will most likely improve in our next revision. Furthermore, the results of Kitaev et al. are those reported in the paper which makes use of an additional factored self-attention layer. Finally, we use the training setting suggested by Kitaev et al. for our experiments which makes this comparison favorable to their parser. Despite all this, our approach still outperforms or comes very close to the Kitaev et al. results while being significantly faster.
> |               | Basque | French | Hebrew | Hungarian | Polish | Swedish |
> | ------------- | ------ | ------ | ------ | --------- | ------ | ------- |
> | Kitaev et al. | 90.70  | 87.35  | **92.95**  | 94.60     | **96.26**  | **89.94**   |
> | Ours          | **90.93**  | **87.59**  | 92.69  | **94.64**     | 95.86  | 89.29   |
>
> **Response to the reproducibility score:**
> We emphasize that we have provided all the hyperparameters and details on the model architectures in Appendix E and Section 4-Model Details. Furthermore, The code will be open-sourced upon publication and will include instructions for reproducing all the experiments.
> Regarding the dataset generation or any other point of confusion, we would be happy to clarify any questions or concerns that Reviewer Fb95 might have in our next revision.

---

### Meta-Review · Area_Chair_9Gdz · 2023-09-18

**Recommendation:** 3

**Metareview:**

This paper investigates the feasibility of Transformers to approximate CKY algorithms. The authors evaluate their Transformer-based  CKY approximations on two data sets: standard benchmarks and a synthetic data set. This paper also introduces some changes in Transformer-based approximation, including a cross-entropy loss over possible constituents for spans, an inductive bias, decoding prediction from gradients etc. The main findings are 1): their Transformer-based  CKY approximation outperforms and runs faster over CKY algorithms on half of the standard benchmark sets; 2) however, experiments on more ambiguous synthetic data suggest their approach is not able to fully capturing the CKY. Those findings are beneficial to our parsing community.

---

### Decision · Program_Chairs · 2023-10-07

**Decision:**

Accept-Findings

**Comment:**

This paper investigates the feasibility of Transformers to approximate CKY algorithms. The authors evaluate their Transformer-based  CKY approximations on two data sets: standard benchmarks and a synthetic data set. This paper also introduces some changes in Transformer-based approximation, including a cross-entropy loss over possible constituents for spans, an inductive bias, decoding prediction from gradients etc. The main findings are 1): their Transformer-based  CKY approximation outperforms and runs faster over CKY algorithms on half of the standard benchmark sets; 2) however, experiments on more ambiguous synthetic data suggest their approach is not able to fully capturing the CKY. Those findings are beneficial to our parsing community.